



# The EUPPBench postprocessing benchmark dataset v1.0

Jonathan Demaeyer[1,2], Jonas Bhend[3], Sebastian Lerch[4], Cristina Primo[5], Bert Van Schaeybroeck[1], Aitor Atencia[6], Zied Ben Bouallègue[7], Jieyu Chen[4], Markus Dabernig[6], Gavin Evans[8], Jana Faganeli Pucer[9], Ben Hooper[8], Nina Horat[4], David Jobst[10], Janko Merše[11], Peter Mlakar[9,11], Annette Möller[12], Olivier Mestre[13], Maxime Taillardat[13], and Stéphane Vannitsem[1,2]

[1]Royal Meteorological Institute of Belgium, Brussels, Belgium
[2]European Meteorological Network (EUMETNET), Brussels, Belgium
[3]Federal Office of Meteorology and Climatology MeteoSwiss, Zurich, Switzerland
[4]Karlsruhe Institute of Technology, Karlsruhe, Germany
[5]Deutscher Wetterdienst, Offenbach, Germany
[6]GeoSphere Austria, Vienna, Austria
[7]European Centre for Medium-Range Weather Forecasts, Reading, United Kingdom
[8]Met Office, Exeter, United Kingdom
[9]University of Ljubljana, Faculty of Computer and Information Science, Slovenia
[10]University of Hildesheim, Hildesheim, Germany
[11]Slovenian Environment Agency, Ljubljana, Slovenia
[12]Bielefeld University, Bielefeld, Germany
[13]Meteo France, Ecole Nationale de la Meteorologie, Toulouse, France

**Correspondence:** Jonathan Demaeyer (Jonathan.Demaeyer@meteo.be)

**Abstract.** Statistical Postprocessing of medium-range weather forecasts is an important component of modern forecasting systems. Since the beginning of modern data science, numerous new postprocessing methods have been proposed, complementing an already very diverse field. However, one of the questions that frequently arises when considering different methods in the framework of implementing operational postprocessing is the relative performance of the methods for a given specific task. It is particularly challenging to find or construct a common comprehensive dataset that can be used to perform such comparisons. Here, we introduce the first version of *EUPPBench*, a dataset of time-aligned forecasts and observations, with the aim to facilitate and standardize this process. This dataset is publicly available at https://github.com/EUPP-benchmark/climetlab-eumetnet-postprocessing-benchmark. We provide examples on how to download and use the data, propose a set of evaluation methods, and perform a first benchmark of several methods for the correction of 2-meter temperature forecasts.

## 1 Introduction

Since the advent of numerical weather prediction, statistical postprocessing techniques have been used to correct forecasts biases and errors. The term "postprocessing techniques" here refers to methods which use past forecasts and observations combined together to learn about the models weather forecast deficiencies, aiming to use that knowledge to correct their future forecasts. Nowadays, postprocessing of weather forecasts is an important part of the forecasting chain in modern prediction systems at national and international meteorological services.



Many postprocessing approaches have been proposed during the last half century, ranging from the so-called *Perfect Prog* method (Klein et al., 1959; Klein and Lewis, 1970) to *Bayesian Model Averaging* (BMA) techniques (Raftery et al., 2005), and including the emblematic *Model Output Statistics* (MOS) approach (Glahn and Lowry, 1972). Some of these methods have been adapted to deal with ensemble forecasts and also calibrate the associated forecast probabilities, like the EMOS method (Gneiting et al., 2005). Recently, machine learning-based methods were proposed (Taillardat et al., 2016; Rasp and Lerch, 2018; Bremnes, 2020), which were shown to improve upon the conventional methods (Schulz and Lerch, 2022).

Systematic intercomparison exercises of both univariate (e.g., Rasp and Lerch, 2018; Schulz and Lerch, 2022; Chapman et al., 2022; Chen et al., 2022c) and multivariate (e.g., Wilks, 2015; Perrone et al., 2020; Lerch et al., 2020; Lakatos et al., 2022) postprocessing methods exist, often based on artificial simulated datasets mimicking properties of real-world ensemble forecasting systems, or based on real-world datasets consisting of ensemble forecasts and observations for specific use-cases. However, there currently is no comprehensive, widely applicable benchmark dataset available for station- and grid-based postprocessing that facilitates re-use and comparisons by serving multiple purposes, e.g. by including a large set of potential input predictors and several target variables relevant to operational weather forecasting at meteorological services. The aim of the present work is to pave the way towards achieving these aims, with the publication of an extensive - *analysis-ready* - forecast and observation dataset, both gridded and at station locations. By an analysis-ready dataset, we mean that the dataset formatting is tailored to obtain the most optimal match between observations and forecasts. In practice, this means that the observations are not provided as conventional time series but rather at the times and locations that match the forecasts.

Recently, the need for a common platform based on which different postprocessing techniques of weather forecasts can be compared, was highlighted (Vannitsem et al., 2021), and extensively discussed in the context of the European Meteorological Network (EUMETNET) working group on postprocessing, called EUPP. Here, we introduce the first step in the development of such a platform, in the form of an easily-accessible dataset that can be used by a large community of users interested in the design of efficient postprocessing algorithms of weather forecasts for different applications. As stated in Dueben et al. (2022), comprehensive benchmark datasets are needed to enable a fair quantitative comparison between different tools and methods, while reducing the need to design and build them, a task which requires domain-specific knowledge. In this view, common benchmark datasets facilitate the collaboration of different communities with different expertise, by lowering the "energy barrier" required to embark on specific problems which would have otherwise required an excessive and discouraging amount of resources.

Many datasets related to weather and climate prediction were released during the last 3 years, emphasizing the need and appetite of the field for ever more data. For instance, datasets have been published related to sea ice drift (Rabault et al., 2022), to hydrology (Han et al., 2022), to learning of cloud classes (Zantedeschi et al., 2019), to sub-seasonal and seasonal weather forecasting (Rasp et al., 2020; Garg et al., 2022; Lenkoski et al., 2022; Wang et al., 2022), and - most relevant to the present work - to the benchmarking of postprocessing methods (Haupt et al., 2021; Ashkboos et al., 2022; Kim et al., 2022). Haupt et al. (2021) distribute a collection of (partly pre-existing) different datasets for specific postprocessing tasks, including ensemble forecasts of the Madden-Julian Oscillation, integrated vapour transport over the Eastern Pacific and Western United





States, temperature over Germany and surface road conditions in the United Kingdom. By contrast, Ashkboos et al. (2022) provide a reduced set of global gridded 10-member ECMWF ensemble forecasts for selected target variables.

Providing weather- or climate-related datasets to the scientific community in a standardized and persistent way remains a challenge, which was recently simplified by the introduction of efficient tools to store and provide data to the users. We can mention for example *xarray* (Hoyer and Joseph, 2017), *Zarr* (Miles et al., 2020), *dask* and the package *climetlab* recently

developed by the European Center for Medium-range Weather Forecasts (ECMWF). The dataset introduced in the present article is for instance provided by a *climetlab plugin*, but also accessible through other means and programming languages (see the Supplementary Information).

The EUPPBench dataset consists of gridded and point ECMWF sub-daily forecasts of all kinds (deterministic high-resolution, ensemble forecasts and reforecasts) over central Europe (see Figure 1). EUPPBench encompasses both station- and grid-based

forecasts for many different variables, enabling a large variety of applications. This - complement by the inclusion of reforecasts - enables a realistic representation of operational postprocessing situations, allowing users and institutions to learn and improve their skills on this crucial process. These operational aspects are, to our knowledge, missing in the currently available postprocessing benchmark datasets. The forecasts and reforecasts of EUPPBench are paired with station observations and gridded reanalysis for the purpose of training and verifying postprocessing methods. To demonstrate how this dataset can be used,

a benchmark of state-of-the-art postprocessing methods has been conducted to improve medium-range temperature forecasts. Although limited in scope, the outcome of this benchmark already emphasizes the potential of the dataset to provide meaningful results and provides useful insights in the potential, diversity and limitations of postprocessing over the study domain. Additionally, performing the benchmark for the first time with a large community also allows to address the usefulness of the established guidelines and protocols and to draw lessons-learned which are important assets for the delivery of many more

benchmarks to come.

This article is structured as follow: The dataset structure and metadata is introduced in Section 2. The design and the verification setup of the benchmark which was carried out on the occasion of the publication of this dataset is explained in Section 3, while in Section 4, the benchmarked methods are detailed. The results of the benchmark are presented in Section 5. We draw some interesting conclusions in Section 6 and give some prospects on the future development of the dataset and of

75 other benchmarks to come. Finally, the code of the benchmark and the data availability of the dataset are provided in Section 7.

## 2 EUPPBench v1.0 dataset

The EUPPBench dataset is available on a portion of Europe covering 47.75° to 53.5° in latitude, and 2.5° to 10.5° in longitude. Therefore, this domain includes mainly Belgium, France, Germany, Switzerland, Austria and the Netherlands. It is stored in Zarr format, a CF compatible format (Gregory, 2003; Eaton et al., 2003) which provides easy access and allows users to

80 "slice" the data along various dimensions in an effortless and efficient manner. In addition, the forecast and observation data are already paired together along corresponding dimensions, providing therefore an analysis-ready dataset for postprocessing benchmarking purposes.

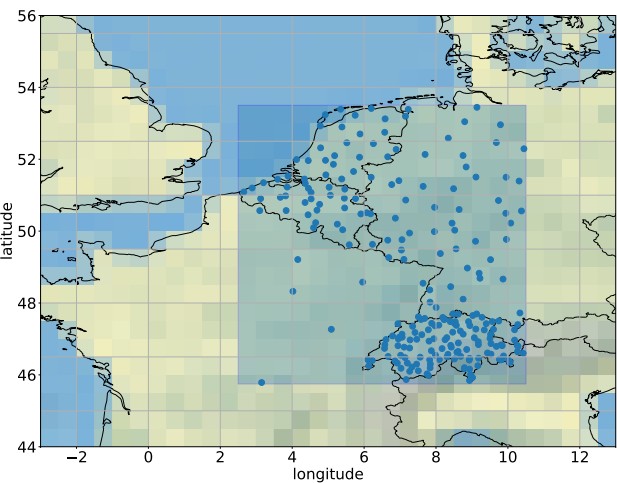

**Figure 1.** The spatial domain of the gridded dataset and the position of the stations included.

EUPPBench includes both the 00Z (midnight) sub-daily ensemble forecasts and reforecasts (Hagedorn et al., 2008; Hamill et al., 2008) produced by the Integrated Forecasting System (IFS) of ECMWF during the years 2017 and 2018, which are released by the forecasting center under the CC-BY-4.0 license. Therefore there are 730 forecast dates and 209 reforecast dates over the 2-years span, reforecasts being produced twice a week (Monday and Thursday). Apart from the ensemble forecasts and reforecasts, the high-resolution deterministic forecasts is also included. Each reforecast date, however, consists of 20 past forecasts computed with the model version valid at the reforecast date, and initialized from 1 to 20 years in the past at the same date of the year, thereby covering the period 1997 - 2017. In total, there are 4180 reforecasts. The number of ensemble members is 51 and 11 for the forecasts and reforecasts, respectively. This includes the forecast control run which is assumed to have the "closest" initial conditions to reality. The choice of the years 2017 and 2018 was motivated by the relative small number of model changes of the ECMWF forecast system during that period, and most importantly, the absence of model resolution modifications, as shown by Table 1. This implementation constraint is crucial to ensure that no supplementary model error biases are introduced in the datasets, as those biases can lead to a more-or-less severe degradation of the postprocessing performances (Lang et al., 2020; Demaeyer and Vannitsem, 2020).

The forecasts and reforecasts time steps are 6-hourly (including the 0-th analysis time steps) up to a lead time of 120 hours (5 days). The variables considered are mainly surface variables and can be classified in two main categories: instantaneous or processed. Table 2 details these two different kinds of variable. Here, a "processed variable" means that the corresponding variable has either been accumulated, averaged or filtered over the past 6 hours. In addition to these surface variables, the Ex-





**Table 1.** ECMWF IFS model changes during the 2017-2018 time span

| Implementation date | Summary of changes | Resolution | Full IFS documentation |
|---|---|---|---|
| 05-Jun-2018 | Cycle 45r1 | Unchanged | Cycle 45r1 full documentation |
| 11-Jul-17 | Cycle 43r3 | Unchanged | Cycle 43r3 full documentation |

Source: https://www.ecmwf.int/en/forecasts/documentation-and-support/changes-ecmwf-model

treme Forecast Indices[1] (Lalaurette, 2003; Zsótér, 2006) and some pressure-level variables are also available (see respectively the Tables 3 and 4).

### 2.1 General data structure

The EUPPBench dataset consists of observations and forecasts in two types: a gridded dataset and a dataset at station locations. All forecasts are based on the ECMWF IFS forecasts. However, while the observational dataset at the station locations is based on ground measurements, the reanalysis ERA5 is taken as the gridded observational dataset. All forecast and reforecast datasets are provided for 31 variables, and additionally, the forecast dataset includes 9 EFI variables. The observations, on the other hand, include only 5 and 21 variables for the station-location and gridded datasets, respectively. Additionally metadata on the model and observations is provided.

How to access the datasets is documented in the Section 7. We now detail in the following subsections the sources and properties of both dataset formats.

### 2.2 Gridded data

All gridded EUPPBench data is provided on a regular grid of 0.25° × 0.25° corresponding roughly to a 25 km horizontal resolution at mid-latitude. As aforementioned, the forecasts and reforecasts are provided by the ECMWF forecasting model in operation at the moment of their issuance. They have both been re-gridded from the ECMWF original ensemble forecasts *O640* (or *O1280* for the deterministic forecasts)[2] grid to the regular grid using the ECMWF MIR interpolation package (Maciel et al., 2017), provided automatically by the MARS archive system. This re-gridding was done to be in line with the resolution of the ERA5 reanalysis (Hersbach et al., 2020) which provides the gridded observations of the EUPPBench dataset.

We recognize that gridded observational datasets over the study domain exist for specific variables that are more accurate than ERA5. For instance, in the case of precipitation-related variables (like the *total precipitation* contained in the dataset at hand), ERA5 has been shown to provide - compared to other datasets - a poor agreement with stations observations (Zandler et al., 2019), mixed performances when used to derive hydrological products (Hafizi and Sorman, 2022), yet, good results when using Perfect prog downscaling methods (Horton, 2022). Notwithstanding, we emphasize that the goal of this gridded dataset is to provide a representative "truth" for the purpose of benchmarking of postprocessing methods. Additionally, the availability

---

[1]Commonly abbreviated as "EFI".

[2]The ensemble forecasts grid *O640* has a horizontal resolution of 18 km while the deterministic forecasts grid *O1280* has a 9 km horizontal resolution.

**Table 2.** List of instantaneous and processed forecast variables on the surface level available in EUPPBench, all available in the EUPPBench gridded and station-location forecast datasets, and the availability of the corresponding gridded and station-location observations.

| Parameter name | Short name | Units | Gridded obs. | Station obs. |
|---|---|---|---|---|
| 2 metre temperature | t2m | K | x | x |
| 10 metre U wind component | 10u | m s$^{-1}$ | x | |
| 10 metre V wind component | 10v | m s$^{-1}$ | x | |
| Total cloud cover | tcc | $\in [0,1]$ | x | x |
| 100 metre U wind component | 100u | m s$^{-1}$ | | |
| 100 metre V wind component | 100v | m s$^{-1}$ | | |
| Convective available potential energy | cape | J kg$^{-1}$ | x | |
| Soil temperature level 1 | stl1 | K | x | |
| Total column water | tcw | kg m$^{-2}$ | x | |
| Total column water vapour | tcwv | kg m$^{-2}$ | x | |
| Volumetric soil water layer 1 | swvl1 | m$^3$ m$^{-3}$ | x | |
| Snow depth | sd | m | x | |
| Convective inhibition | cin | J kg$^{-1}$ | | |
| Visibility | vis | m | | x |
| Total precipitation | tp6 | m | x | x |
| Surface sensible heat flux | sshf6 | J m$^{-2}$ | x | |
| Surface latent heat flux | slhf6 | J m$^{-2}$ | x | |
| Surface net solar radiation | ssr6 | J m$^{-2}$ | x | |
| Surface net thermal radiation | str6 | J m$^{-2}$ | x | |
| Convective precipitation | cp6 | m | x | |
| Maximum temperature at 2 metres | mx2t6 | K | x | |
| Minimum temperature at 2 metres | mn2t6 | K | x | |
| Surface solar radiation downwards | ssrd6 | J m$^{-2}$ | x | |
| Surface thermal radiation downwards | strd6 | J m$^{-2}$ | x | |
| 10 metre wind gust | 10fg6 | m s$^{-1}$ | x | x |

Remark: A '6' was added to the usual ECMWF short names to indicate the span (in hours) of the accumulation or filtering.

of a wide range of variables in ERA5 and the spatio-temporal consistency among different meteorological variables are very
important in this context cannot be provided by gridded observational datasets.



**Table 3.** List of available Extreme Forecast Indices, all available in the EUPPBench gridded and station-location forecast datasets.

| Parameter name | Short name |
|---|---|
| 2 metre temperature EFI | 2ti |
| 10 metre wind speed EFI | 10wsi |
| 10 metre wind gust EFI | 10fgi |
| CAPE EFI | capei |
| CAPE shear EFI | capesi |
| Maximum temperature at 2m EFI | mx2ti |
| Minimum temperature at 2m EFI | mn2ti |
| Snowfall EFI | sfi |
| Total precipitation EFI | tpi |

Remark: By definition, observations are not available for the EFI. The EFI are available for the model step ranges (in hours) 0-24, 24-48, 48-72, 72-96, 96-120, 120-144 and 144-168. Values range of EFI is from -1 to +1.

**Table 4.** List of variables on pressure levels, all available in the EUPPBench gridded and station-location forecast datasets.

| Parameter name | Level | Short name | Units |
|---|---|---|---|
| Temperature | 850 | t | K |
| U component of wind | 700 | u | $m\ s^{-1}$ |
| V component of wind | 700 | v | $m\ s^{-1}$ |
| Geopotential | 500 | z | $m^2\ s^{-2}$ |
| Specific humidity | 700 | q | $kg\ kg^{-1}$ |
| Relative humidity | 850 | r | % |

Remark: Only gridded observations (reanalysis) are available for these variables.

## 2.3 EUPPBench data at station locations

Subdaily station observations have been provided by many National Meteorological Services (NMS) participating in the construction of this dataset, and a big part of the station data can be considered as open data (see section 7). The observations of 234 stations cover the entire 22-year time period 1997-2018 necessary to match the reforecasts and forecasts. The elevation of these stations varies from a few meters below the sea level up to 3562 meters for the Jungfraujoch station in Switzerland. These stations constitute the most authoritative sources of information about weather and climate provided in each of the involved countries, being constantly monitored and the quality of the data being checked.



**Table 5.** List of available constant fields

| Parameter name | Short name | Units |
|---|---|---|
| Land use | landu | 1,2,3,...,44,48 |
| Model terrain height | mterh | m |
| Surface Geopotential | z | $m^2\,s^{-2}$ |

The land usage is extracted from the CORINE2018 dataset (Copernicus
Land Monitoring Service, 2018). More details are provided in the
"legend" entry of the metadata within each file.
The model terrain height is extracted from the EU-DEM v1.1 data
elevation model dataset (Copernicus Land Monitoring Service).
Finally, the model orography can be obtained by dividing the surface
geopotential by $g = 9.80665\,\mathrm{m\,s^{-2}}$.

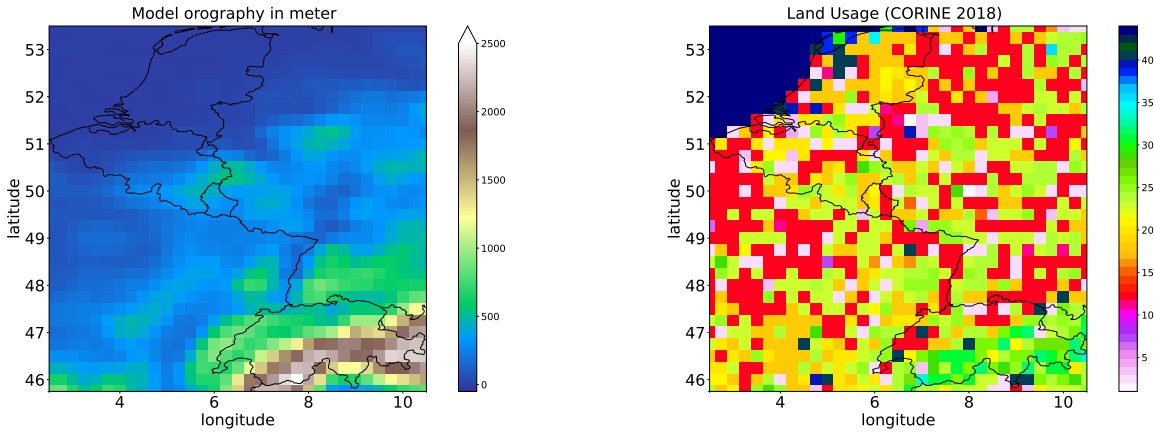

**Figure 2.** Static fields in the gridded dataset. Left panel: The model orography obtained by dividing the model surface geopotential by $g = 9.80665\,\mathrm{m\,s^{-2}}$. Right panel: Grid point land usage provided by the CORINE 2018 dataset (Copernicus Land Monitoring Service, 2018). Numerical codes indicating the usage categories is included in the dataset metadata.

The EUPPBench dataset at station locations consists of the ECMWF forecasts and reforecasts at the grid point closest to the station locations and the associated observations, matched for each lead time. As shown in the Table 2, there are mainly 5 variables currently available: 2 metre temperature (*t2m*), total cloud cover (*tcc*), visibility (*vis*), total precipitation (*tp6*) and 10 metre wind gust (*10fg6*). More observation variables will be added in subsequent versions of the dataset.

## 2.4 Static data and metadata

In addition to the forecasts and reforecasts, auxiliary fields are provided, such as the land usage or the surface geopotential which is proportional to the model orography (see Figure 2). Table 5 synthesizes this part of the dataset. These constant fields have been extracted and are also provided in the stations metadata.



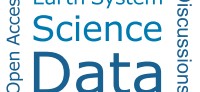

**Table 6.** The metadata provided in the files of the gridded forecasts, reforecasts and observational datasets.

| Metadata | Description |
|---|---|
| **latitude** | Latitude of the grid points. |
| **longitude** | Longitude of the grid points. |
| **depthBelowLandLayer** | Layer below the surface (valid for some variables only, here there is only the upper surface level). |
| **number** | Number of the ensemble member. The 0-th member is the control run. Also present in observation for compatibility reasons, but set to 0. |
| **time** | Forecast or reforecast date (reforecasts are only issued on Mondays and Thursdays). |
| **year** | Dimension to identify the year in the past, year=1 means a forecast valid 20 years ago at the reforecast day and month, year=20 means a forecast valid one year before the reforecast date. Only valid for reforecasts. |
| **step** | Step of the forecast (the lead time). |
| **surface** | Layer of the variable considered (here there is just one, at the surface). |
| **isobaricInhPa** | Pressure level in hectopascal (or millibar). |
| valid_time | Actual time and date of the corresponding forecast data. |

Remark: **Bold** metadata denotes dimensions indexing the datasets.

Depending on the kind of dataset, dimensions and different information are embedded in the data: For gridded data, the metadata available in the forecast, reforecast and observation datasets are detailed in Table 6. For station data, the forecast and reforecast metadata are detailed are detailed in Table 7, while the observation metadata are detailed in Table 8. For all data, attributes specifying the sources and the license are always present.

## 3 Postprocessing benchmark

To illustrate the usefulness of the EUPPBench dataset, a benchmark of several state-of-the-art postprocessing methods - and many of which are currently in operation in NMSs - was performed, along with some more recent and more advanced methods. This first exercise was based on a small subset of the dataset. Along the same line, the verification process of this benchmark also focused on some general aspects typically considered for operational postprocessing. In this section, we describe the general framework that we used to conduct this benchmark. The following sections will be devoted to the methods and the results obtained. We begin by detailing the design of this experiment.

### 3.1 Experiment design

The postprocessing benchmark at hand considers the correction of the ensemble forecasts of the 2 meter temperature at the nearest forecast grid point from every station available in the dataset, spanning several European countries and the whole





**Table 7.** The metadata provided in the files of the forecast, reforecast at the station locations.

| Metadata | Description |
|---|---|
| station_latitude | Latitude of the station. |
| station_longitude | Longitude of the station. |
| station_altitude | Elevation of the station (in meter). |
| **station_id** | Unique identifier of the station. |
| **depthBelowLandLayer** | Layer below the surface (valid for some variables only, here there is only the upper surface level). |
| **number** | Number of the ensemble member. The 0-th member is the control run. Also present in observation for compatibility reasons, but set to 0. |
| **time** | Forecast or reforecast date (reforecasts are only issued on Mondays and Thursdays). |
| **year** | Dimension to identify the year in the past, year=1 means a forecast valid 20 years ago at the reforecast day and month, year=20 means a forecast valid one year before the reforecast date. Only valid for reforecasts. |
| **step** | Step of the forecast (the lead time). |
| **surface** | Layer of the variable considered (here there is just one, at the surface). |
| **isobaricInhPa** | Pressure level in hectopascal (or millibar). |
| station_land_usage | Land usage at the station location, extracted from the CORINE 2018 dataset. |
| station_name | Name of the station. |
| model_latitude | Latitude of the model grid point. |
| model_longitude | Longitude of the model grid point. |
| model_altitude | True elevation (in meter) of the model grid point, extracted from the EU-DEMv1.1 data elevation model dataset. |
| model_orography | Surface height (in meter) in the model at the model grid point. |
| model_land_usage | Land usage at the model grid point, extracted from the CORINE 2018 dataset. |
| valid_time | Actual time and date of the corresponding forecast data. |

Remark 1: **Bold** metadata denotes dimensions indexing the datasets.

Remark 2: The metadata with 'model' in their name indicate properties of the model grid point the closest to the station location, and at which the forecasts corresponding to the station observations was extracted from the gridded dataset.

EUPPBench area. We note that this area includes orographically difficult regions, nearly-flat plains, and also stations close to the sea or located on islands. Discrepancies may therefore occur between forecast and observations due to poor observational representativity at the scale of the model or to challenges in the model representation of a wide range of physical processes. We note also that due to the coarse nature of the gridded forecast dataset, the forecast grid points are not evenly situated with respect to the stations they represent, with sometimes huge differences in elevation or situation (e.g. forecast point at sea), that 160 may induce large temperature biases.





**Table 8.** Station observations metadata

| Metadata | Description |
|---|---|
| altitude | Elevation of the station (in meter). |
| land_usage | Land usage at the station location, extracted from the CORINE 2018 dataset. |
| latitude | Latitude of the station. |
| longitude | Longitude of the station. |
| **station_id** | Unique identifier of the station. |
| station_name | Name of the station. |
| **step** | Step of the forecast (the lead time). |
| **time** | Forecast or reforecast date (reforecasts are only issued on Mondays and Thursdays). |

Remark: **Bold** metadata denotes dimensions indexing the datasets.

Within this simplified benchmark exercise, the only predictor that could be used to perform the postprocessing is the temperature at 2 meter itself. Additionally the use of the (static) metadata was allowed and some methods used latitude and longitude, elevation, land use, model orography, and also the day of the year. The 11-member reforecasts produced during the 2017-2018 period were considered as training data while the 51-member forecasts for the same period was used as test data for verification.
This setup introduced some challenges for the implementation of some of the postprocessing methods described below.

To avoid a potential overlap between the reforecasts and the forecasts, the forecast from 2017 that were included in the reforecasts of 2018 have been removed from the training dataset. Since the ECMWF reforecasts date are being produced each Monday and Thursday, the reforecasts (per lead time) for 2017 do not overlap with those of 2018.

However, one notable difference between the training data and the test data is the number of ensemble members: ensemble
forecasts contain 51 members while ensemble reforecasts include 11 members. Also, note that in both cases, the ECMWF control run forecast is included in the ensemble (as the 0-th ensemble member). The high-resolution deterministic forecast runs were not used nor postprocessed in the current benchmark.

## 3.2 Verification setup and methodology

Forecasts are given at a particular run time for a particular hour and place of interest. Differences between forecast and ob-
175 servation provide an idea about the forecast error. When the study extends to time periods and areas, these errors have to be aggregated, and according to how the aggregation is done, the analyses will focus on different aspects of the forecasts: Can we distinguish spatial patterns or dependence on the elevation in skill? How does the forecast quality decrease with the lead time? Are there systematic deviations between the forecast members and the observations. In addition, when dealing with an ensemble of forecasts, is the ensemble well calibrated, i.e. is the forecast spread a good measure of the uncertainty?
The verification study answers these questions by comparing the performance of the different methods showing aggregated statistical scores using temporal series, maps and rank histograms. More specifically, the post-processed forecasts at the station





locations within the test dataset (2017-2018) are compared here with observations. We use two metrics that quantify the forecast quality, or in other words, how well the ensemble forecast matches the observations: the Bias and the Continuous Ranked Probability Score (CRPS) Hersbach (2000). Bias addresses the skill of the ensemble average and corresponds to the

185 average differences between forecast and observation. CRPS, on the other hand, addresses the probabilistic skill by comparing the Cumulative Distribution Functions (CDF) of the forecast against the corresponding observation. The calibration of the ensemble is analyzed via a spread/skill score equal to the ratio of the average ensemble standard deviation divided by the root-mean squared error of the ensemble mean. As a reference forecasting dataset the raw IFS ensemble at the nearest forecast grid point from every station is used with an additional lapse-rate correction of 6.5°C/km to account for the difference between the

190 IFS orography and the station elevation. Some results are also presented conditioned on the station elevation to detect remnant orographic influences.

The verification results for the different postprocessing methods are obtained after performing quality-control tests on the initial data to detect for possible inconsistencies, unrealistic values and missing data. Missing postprocessed predictions for individual time steps and locations in the test set are replaced by the direct model output (DMO). Postprocessing methods with

195 missing values are therefore intentionally penalized. The rationale behind this is that EUPP aims for improving *operational* forecasting systems in which forecasts need to be provided in any case. Additionally, this approach discourages hedging, i.e. artificially increasing the performance of a postprocessing method, by replacing known cases with underperforming skill by a missing value. Moreover, significant tests are run to assess if the score differences were significant.

## 4 Postprocessing methods

Along with the dataset and verification framework described above, the present work further includes a collection of forecasts of exemplary postprocessing methods along with corresponding code for their implementation. Note that with providing forecasts of a selected set of methods, we do not intend to provide a comprehensive or systematic comparison to establish the "best" approach, but rather aim to present an overview of both commonly used and more advanced methods ranging from approaches from statistics to machine learning. Those can be used in subsequent research for developing extensions to existing approaches

and for comparing novel methods to established baselines. Short descriptions of the methods available in the present benchmark are provided below, verification results are presented in Section 5. Specific details regarding the adaptation and implementation of the different methods, as well as code are available from the corresponding Github repositories[3]. For a general overview of recent developments in postprocessing methodology, we refer to Vannitsem et al. (2018, 2021).

Within the present section, we use the following notation. For a specific forecast instance $t$ (at a specific location and for a

210 specific initialization and lead time), we denote the ensemble forecasts by $x_m(t), m = 1, ..., M$, their mean value by $\mu^{\text{ens}}(t)$, and their standard deviation by $\sigma^{\text{ens}}(t)$. The corresponding observation is denoted by $y(t)$.

---

[3]See a detailed list of the methods Github repositories at https://github.com/EUPP-benchmark/ESSD-benchmark.



## 4.1 Accounting for systematic and representativeness errors (ASRE)

ASRE postprocessing tackles systematic and representativeness errors in two independent steps. A local bias correction approach is applied to correct for systematic errors. For each station and each lead time, the averaged difference between re-
forecasts and observations in the training dataset is computed and removed from the forecast in the validation dataset. The difference averaging is performed over all training dates centered around the forecast validity date within a window of $\pm$ 30 days.

Representativeness errors are accounted for separately using a universal method inspired by the *Perfect Prog* approach (Klein et al., 1959; Klein and Lewis, 1970). A normal distribution is used to represent the diversity of temperature values that can be
observed at a point within an area given the average temperature of that area. For an area of a given size (i.e., a model grid box), the variance of the distribution is expressed as a function of the difference between station elevation and model orography only (see Eq. 4 in Ben Bouallègue, 2020). Random draws from this probability distribution is added to each ensemble member to simulate representativeness uncertainty.

## 4.2 Reliability Calibration (RC)

Reliability Calibration is a simple, non-parametric technique that specifically targets improving the forecast reliability without degrading forecast resolution. Two additional steps are applied prior to Reliability Calibration, targeted at correcting forecast bias; initially a lapse rate correction of 6.5°C/km between the station elevation and model orography is applied, followed by a simple bias correction calculated independently at each station. Following bias correction, probabilistic forecasts are derived from the bias corrected ensemble member forecasts by calculating the proportion of ensemble members which exceed
thresholds at 0.5°C intervals. At each threshold, the exceedance probabilities are calibrated separately. The Reliability Calibration implementation largely follows Flowerdew (2014), although in this study, all sites are aggregated into a single reliability table which is used to calibrate forecasts across all sites. As in Flowerdew (2014), a set of equally spaced percentiles are extracted using linear interpolation between the thresholds, which are treated as pseudo-ensemble members for verification. The non-parametric nature of Reliability Calibration makes it attractive for a range of diagnostics, including temperature, if
combined with other simple calibration techniques such as those applied here. Reliability Calibration was implemented using IMPROVER (Roberts et al., 2022), an open-source codebase developed by the Met Office and collaborators.

## 4.3 Member-By-Member postprocessing (MBM)

The Member-By-Member approach calibrates the ensemble forecasts by correcting the systematic biases in the ensemble mean with a linear regression-based MOS technique and rescaling the ensemble members around the corrected ensemble
mean (Van Schaeybroeck and Vannitsem, 2015). This procedure estimates the coefficients $\alpha^{\mathrm{MBM}}$, $\beta^{\mathrm{MBM}}$ and $\gamma^{\mathrm{MBM}}$ in the formula

$$T_m^C(t) = \alpha^{\mathrm{MBM}}(t) + \beta^{\mathrm{MBM}}(t)\,\mu^{\mathrm{ens}}(t) + \gamma^{\mathrm{MBM}}(t)\,\bar{T}_m^{\mathrm{ens}}(t), \tag{1}$$

by optimizing the CRPS, separately for each station and for each lead time. $\bar{T}_m^{\text{ens}}(t) = T_m^{\text{ens}}(t) - \mu^{\text{ens}}(t)$ here denotes the deviation of the member $m$ from the ensemble mean. The results were obtained with the Pythie package (Demaeyer, 2022a),

training on the 11 members of the training dataset to obtain the coefficients $\alpha^{\text{MBM}}$, $\beta^{\text{MBM}}$ and $\gamma^{\text{MBM}}$, and then using them to correct the 51 members forecasts of the test dataset. One of the main advantages of MBM postprocessing is that - by design - it preserves correlations in the forecasts.

### 4.4  Ensemble model output statistics (EMOS)

EMOS is a parametric postprocessing method introduced in Gneiting et al. (2005). The temperature observations are modelled

by a Gaussian distribution. The location ($\mu$) and scale ($\sigma$) parameters of the forecast distribution can be described by two linear regression equations via

$$
y(t) \sim \mathcal{N}(\mu, \sigma) \begin{cases} \mu(t) = \beta_0^{\text{EMOS}} + f_1^{\text{EMOS}}(\text{doy}) + \beta_1^{\text{EMOS}} \mu^{\text{ens}}(t) \\ \log(\sigma) = \gamma_0^{\text{EMOS}} + g_1^{\text{EMOS}}(\text{doy}) + \gamma_1^{\text{EMOS}} \log(\sigma^{\text{ens}}(t)), \end{cases} \tag{2}
$$

with $\beta_0^{\text{EMOS}}$, $\gamma_0^{\text{EMOS}}$, $\beta_1^{\text{EMOS}}$ and $\gamma_1^{\text{EMOS}}$ as regression coefficients, and $f_1^{\text{EMOS}}(\text{doy})$ and $g_1^{\text{EMOS}}(\text{doy})$ as seasonal smoothing functions to capture a seasonal bias of location and scale. The seasonal smoothing function is a combination of annual and bi-

annual base functions ($\sin(2\pi \, \text{doy}/365)$, $\cos(2\pi \, \text{doy}/365)$, $\sin(4\pi \, \text{doy}/365)$, and, $\cos(4\pi \, \text{doy}/365)$) as presented in Dabernig et al. (2017). The implemented EMOS version is based on the R-package `crch` (Messner et al., 2016) with maximum likelihood estimation. 51 equidistant quantiles between 1 and 99 % of the distribution are drawn to match the amount of members from the raw ECMWF forecasts, which were needed for verification. EMOS is applied separately to every station and lead time.

### 4.5  EMOS with heteroscedastic autoregressive error adjustments (AR-EMOS)

AR-EMOS extends the EMOS approach by estimating parameters of the predictive distribution based on ensemble forecasts adjusted for autoregressive behavior (Möller and Groß, 2016). For each ensemble forecast $x_m(t)$, the respective error series $z_m(t) := y(t) - x_m(t)$ is defined and an autoregressive (AR) process of order $p_m$ is fitted to each $z_m(t)$ individually. Based on the estimated parameters of the AR($p_m$) processes an AR-adjusted forecast ensemble is obtained via

$$
\widetilde{x}_m(t) = x_m(t) + \alpha_m^{\text{AR}} + \sum_{j=1}^{p_m} \beta_{m,j}^{\text{AR}} [y(t-j) - x_m(t-j) - \alpha_m^{\text{AR}}], \tag{3}
$$

where $\alpha_m^{\text{AR}}$ and $\beta_{m,j}^{\text{AR}}$, $j = 1, \ldots, p_m$ are the coefficients of the respective AR($p_m$) process. The adjusted ensemble forecasts are employed to estimate the mean and variance parameter of the predictive Gaussian distribution. Estimation of the predictive variance was further refined in Möller and Groß (2020). The method is implemented in the R package `ensAR` (Groß and Möller, 2019). However, some adaptations had to be made to the method and implementation in order to accommodate the

benchmark data, see code documentation in the corresponding Github repository.





### 4.6 D-vine copula based postprocessing (DVQR)

In the D-vine (drawable vine) copula based postprocessing, a multivariate conditional copula $C$ is estimated using a pair-copula construction for the graphical D-vine structure according to Kraus and Czado (2017). D-vine copulas enable a flexible modelling of the dependence structure between the observation $y$ and the ensemble forecast $x_1, \ldots, x_m$ (see, e.g., Möller et al., 2018). The covariates $x_1, \ldots, x_m$ are selected by their predictive strength based on the conditional log-likelihood. Afterwards, D-vine copula quantile regression (DVQR) allows to predict quantiles $\alpha \in (0, 1)$ that represent the postprocessed forecasts via

$$F_{y|x_1,\ldots,x_m}^{-1}(\alpha|x_1(t),\ldots,x_m(t)) := F_y^{-1}\left(C^{-1}(\alpha|F_{x_1}(x_1(t)),\ldots,F_{x_m}(x_m(t)))\right), \tag{4}$$

where $F_{x_i}$ denotes the marginal distributions of $x_i$ for all $i = 1, \ldots, m$, $F_y^{-1}$ the inverse marginal distribution of $y$ and $C^{-1}$ is the conditional copula quantile function. The implementation of this method is mainly based on the R package `vinereg` by Nagler (2020), where the marginal distributions are kernel density estimates. DVQR is estimated separately for every station and lead time using a seasonal adaptive training period.

### 4.7 Distributional regression network (DRN)

Rasp and Lerch (2018) first proposed the use of neural networks (NNs) for probabilistic ensemble postprocessing. In a nutshell, their DRN approach extends the EMOS framework by replacing pre-specified link functions with a NN connecting inputs and distribution parameters, enabling flexible nonlinear dependencies to be learned in a data-driven way. The parameters of a suitable parametric distribution are obtained as the output of the NN, which may utilize arbitrary predictors as inputs, including additional meteorological variables from the NWP system and station information. In our implementation for EUPPBench, we closely follow Rasp and Lerch (2018), and assume a Gaussian predictive distribution. We fit a single DRN model per lead time jointly for all stations, and encode the station identifier and land-use via embedding layers to make the model locally adaptive. Since the use of additional input information has been a key aspect in the substantial improvements of DRN and subsequent extensions in other NN-based methods over EMOS, similar benefits are less likely here due to the limitation to ensemble predictions of the target variable only in the experimental setup, see Rasp and Lerch (2018) for more detailed comparisons.

### 4.8 ANET

ANET is a NN approach, similar to DRN, for postprocessing ensembles with variable member counts. ANET estimates the parameters of a predictive Gaussian distribution jointly for all lead times and over all stations. ANET processes individual ensemble members first, and combines them into a single output inside the architecture later. A dynamic attention mechanism facilitates focusing on important sample members, enabling ANET to retain more information about individual members in cases where the ensemble describes a more complex distribution. Likewise, we take advantage of the fact that we are predicting the parameters of a Gaussian distribution by computing the mean and spread of the residuals $\mu_\Delta^i$ and $\sigma_\Delta^i$ rather than the direct distribution parameter values. ANET thus computes the distribution parameters for a lead time $i$ as $\mu_i^{\text{ANET}}(t) = \mu_i^{\text{ens}}(t) + \mu_{\Delta,i}$, $\sigma_i^{\text{ANET}}(t) = S(\sigma_i^{\text{ens}}(t) + \sigma_\Delta, i)$, where $S$ denotes the softplus activation function $S(x) = \ln(1 + e^x)$, ensuring that the standard



deviation remains positive. To further increase the stability of ANET we randomly varied the number of ensemble members passed to the network during training. The model is trained by minimizing the negative log-likelihood function.

## 5    Results

Here we present the results from the verification of the submission to the benchmark. The CRPS (Fig. 3a) as a measure of forecast accuracy clearly demonstrates the benefit of postprocessing. The elevation-corrected ECMWF DMO exhibits pronounced diurnal variability in CRPS with forecast errors at night being considerably more pronounced than during the day. Postprocessing achieves a reduction of these forecast errors by up to 50% early in the forecast lead time and by 10-40% on day 5. Most postprocessing methods perform similarly with the notable exception of ANET that achieves the lowest CRPS and

exhibits much less diurnal variability in forecast errors.

Postprocessing improves forecast performance by reducing systematic biases (Fig. 3b) and by increasing ensemble spread (Fig. 3c) to account for sources of variability not included in the NWP system. Again the ensemble spread of most postprocessing methods is similar with the notable exception of RC that generates much more dispersed forecasts in particular early in the forecast lead time.

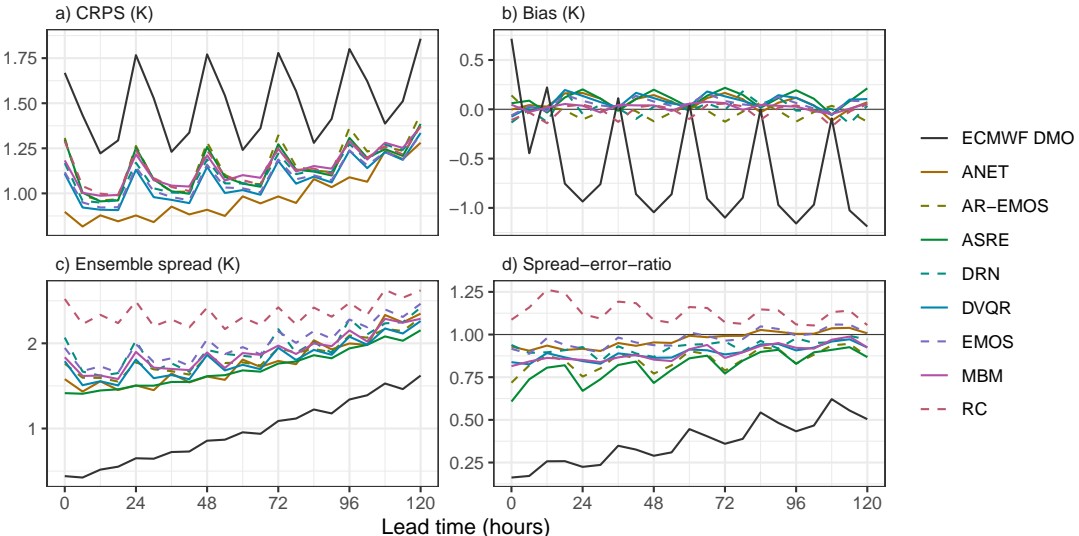

**Figure 3.** Average scores in dependence of lead time.

Forecast calibration is assessed with the spread-error-ratio (Fig. 3d) and the rank histogram (Fig. 4). ECMWF DMO is heavily over-confident i.e. a spread-error-ratio smaller than 1 and a U-shaped rank histogram. Postprocessed forecasts are much better calibrated with indication of some remaining forecast over-confidence for all methods but RC (Fig. 3d). The rank histogram in Fig. 4 allows for a different perspective on forecast calibration with indication of forecast over-dispersion (inverse U-shape) for many of the postprocessed forecasts.





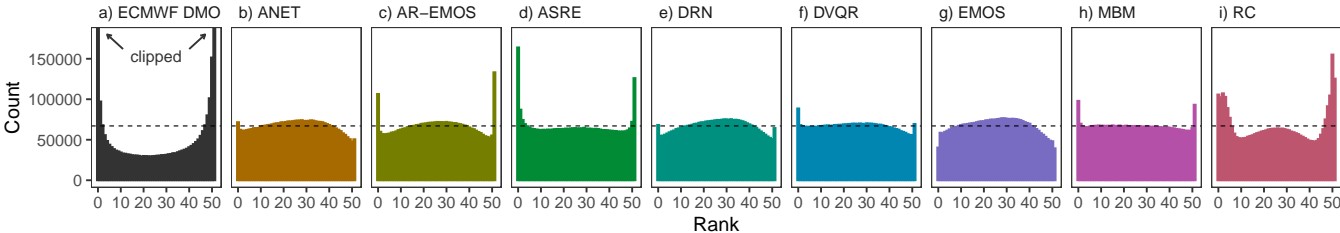

**Figure 4.** Rank histogram of forecasts submitted to the benchmark experiment. Please note that the visualization for ECMWF DMO is clipped for better comparison with the rank histograms of the postprocessed forecasts.

Postprocessed forecasts have been produced for a number of stations in central Western Europe. With very few exceptions, postprocessing improves forecast quality everywhere as illustrated by the positive values of CRPSS in Figure 5. Most of the postprocessing methods perform similarly with more pronounced improvements in complex topography and less pronounced improvements the northern and pre-dominantly flat part of the domain. As a notable exception, ANET forecasts perform better in particular for high-altitude stations in Switzerland and for coastal stations in Belgium and the Netherlands.

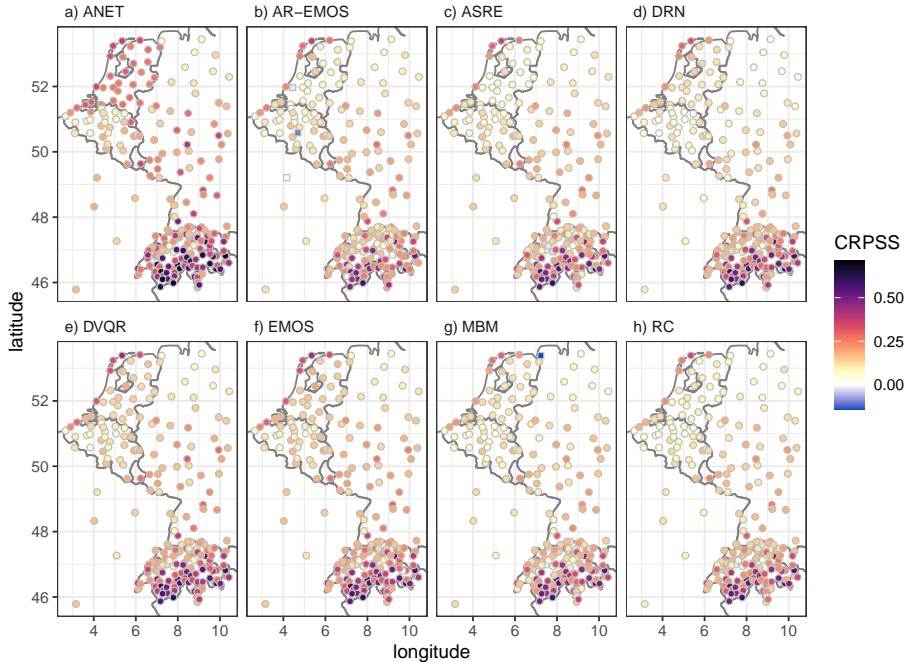

**Figure 5.** Continuous ranked probability skill score (CRPSS) per station. CRPSS is computed using the ECMWF DMO as the reference forecast and positive values indicate that the postprocessed forecasts outperform ECMWF DMO. Stations at which forecast skill is negative are marked by square symbols.





In Fig. 6 we present a range of scores in dependence of the station altitude to further explore the specifics of the postprocessing methods. For example forecasts with the elevation-corrected ECMWF DMO for high-altitude stations are systematically too cold, indicating that the constant lapse rate correction applied to ECMWF DMO is an approximation at best. The AR-EMOS methods appears to produce the smallest biases overall, whereas there is some remaining negative bias at altitude in many of the methods and positive biases in RC. The remaining large biases in the AR-EMOS and EMOS methods are from
missing predictions that have been filled with ECMWF DMO.

The CRPS at each station in Fig. 6 shows that the reduction in forecast errors and correspondingly increased forecast skill is generally more pronounced at altitude. Compared with the other postprocessing approaches, ANET achieves lower CRPS for high-altitude stations (above 1000 m) and for a cluster of stations below 100 m predominantly located in the Netherlands (see also Fig. 5).

The spread-error-ratio as a necessary condition for forecast calibration also reveals considerable differences between postprocessing approaches. The ASRE and RC methods in particular exhibit large variations in spread-error-ratio from station to station, whereas the other methods exhibit much more uniform spread-error-ratios. More detailed analyses of the results would of course be possible, but are beyond the scope of this publication on the benchmark dataset, and will be the subject of a dedicated work.

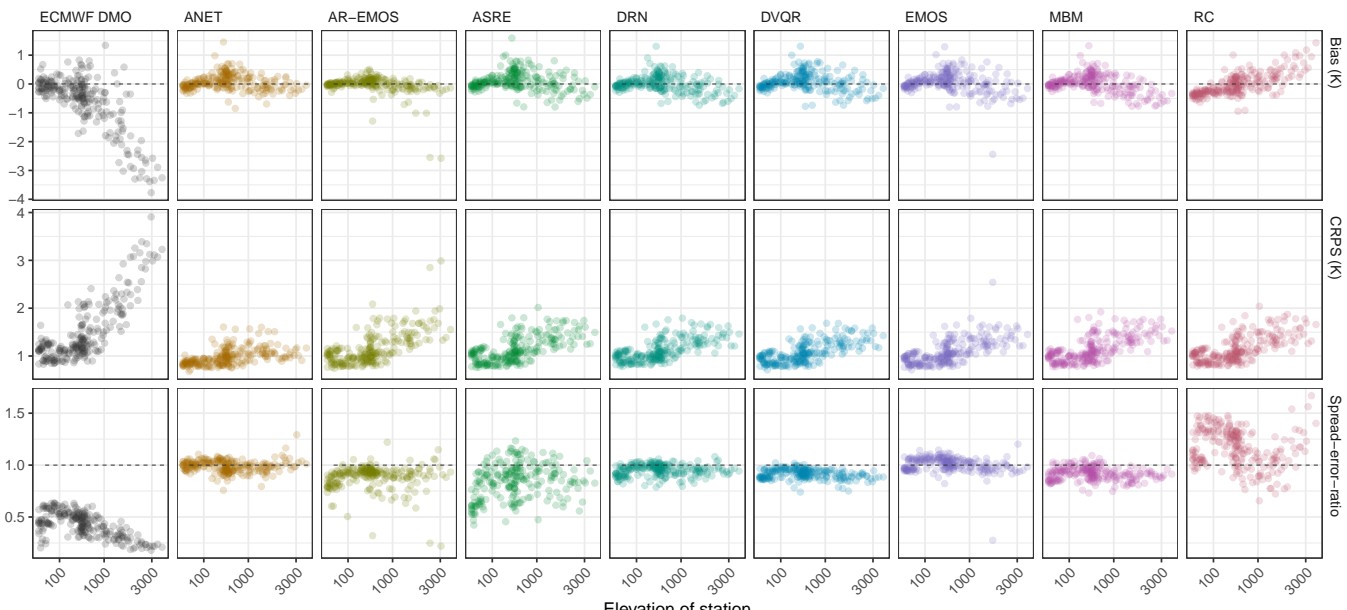

**Figure 6.** Average scores by station in dependence of station elevation, the elevation-corrected ECMWF DMO is shown alongside the results from the postprocessing methods submitted to the benchmark experiment.





## 6 Conclusions and prospects

A benchmark dataset is proposed in the context of the EUMETNET PostProcessing (EUPP) program for comparing statistical postprocessing techniques that are nowadays an integral part of many operational weather-forecasting suites. This dataset includes ensemble forecasts and reforecasts of the ECMWF for the period 2017-2018, and the corresponding gridded and station observations, over a region covering a small portion of western Europe. This region covers a variety of topographies including

345 as coastal, flat and mountainous areas. To illustrate the usefulness of this dataset, a standardized excercise is established in order to allow an objective and rigorous intercomparison of postprocessing methods. This exercise included the contribution of many well-established state-of-the-art postprocessing techniques. Despite the limited scope of the presented exercise, this collaborative effort will serve as a reference framework and will be strongly extended. The whole process includes: (i) the download of the data or their access on the European Weather Cloud (where the dataset is stored, see Section 7); (ii) the ap-

350 plication of the different techniques by the contributors; and (iii) the verification of the results by the verification team. This proof-of-concept proved to be very successful.

While the authors constructed and performed this benchmark, some lessons were learned along the way:

- As much as possible, avoid to maintain an archive or a database of scores for the experiments. Instead compute verification results for the experiments on the fly and only store the summary results. This has the advantage that you can easily
add (or remove) scores, summaries of scores, without going through the complex process necessary to update an archive (with new submissions and additional scores).

- Either be very strict about the format of submitted predictions, or use software that is aware of the NetCDF data model and that can handle slight inconsistencies (e.g. re-ordering of dimensions or dimension values)

- Quality control is imperative: while the verification results generally quickly indicate whether there are any major issues
with the submitted predictions, issues may already arise earlier than that (making a verification impossible). Catching these errors and establishing a feedback loop with the submitters is important. One way to solve this with NetCDF format is to check the NetCDF header of the submission for format compliance.

These points are important for the next EUPP projects, which will aim to harness the full potential of this dataset, by postprocessing other, less predictable variables (e.g. rainfall, radiation), on station and gridded data, and by allowing many

predictors instead of only the target variable itself, as it is the case in the setup of the methods presented here. By considering broader aspects of forecasting (e.g. spatial and temporal aspects) as well as more specific scores, the verification task for these forthcoming studies will allow us to use more advanced and cutting-edge concepts in the field. The lessons learned from these experiments will also be valuable to other groups engaging with the design and operation of such benchmarking experiments. Ultimately, one of the long-term goal of the EUPP module is to provide an automated procedure to upload and compare new

approaches to the existing pool of methods available. It is an ambitious goal, with many challenges ahead, but the benefits it will bring make it worth pursuing.



# 7   Code and data availability

The most straightforward way to access the dataset is through the climetlab EUMETNET postprocessing benchmark plugin at https://github.com/EUPP-benchmark/climetlab-eumetnet-postprocessing-benchmark. This plugin provides an easy access to the dataset stored on the ECMWF European Weather Cloud. An example on how to use the plugin is documented in the Supplementary Information, along with other unofficial ways to access the data.

In addition, the dataset has been stored in Zarr format on Zenodo to preserve it for the long time. See Demaeyer (2022c) for the gridded data and Bhend et al. (2022) for the station data.

However, the Switzerland station data which are part of the dataset are not presently freely available. These station data may be obtained from IDAWEB (https://gate.meteoswiss.ch/idaweb/) at MeteoSwiss and we are not entitled to provide it online. Registration with IDAWEB can be initiated here: https://gate.meteoswiss.ch/idaweb/prepareRegistration.do. For more information, please also read https://gate.meteoswiss.ch/idaweb/more.do?language=en.

The documentation of the dataset is available at https://eupp-benchmark.github.io/EUPPBench-doc/files/EUPPBench_datasets.html and is also provided in the supplementary information.

The various codes and scripts used to perform the benchmark are available on GitHub and have been centralized in a single repository: https://github.com/EUPP-benchmark/ESSD-benchmark-codes. This repository contains links to the scripts sub-repositories along with a detailed description of each method. In addition, these codes have been also uploaded to Zenodo: verification code (Primo-Ramos et al., 2022), MBM method code (Demaeyer, 2022b), Reliability Calibration method code (Evans and Hooper, 2022), ASRE method (Ben Bouallègue, 2022), EMOS method (Dabernig, 2022), AR-EMOS method (Möller, 2022), DRN method (Chen et al., 2022b), DVQR method (Jobst, 2022), ANET method (Mlakar et al., 2022).

Finally, to allow further studies and a better reproducibility, the output data (the corrected forecasts) provided by each methods have also been uploaded to Zenodo (See Chen et al. (2022a)).

*Author contributions.* **Jonathan Demaeyer** lead the overall coordination of the benchmark, as well as the collection and dissemination of the dataset. He further contributed to the verification setup, implemented the MBM method (Section 4.3), and coordinated the writing of the manuscript.

**Jieyu Chen and Nina Horat** implemented the DRN method (Section 4.7).

**Sebastian Lerch** coordinated the writing of Section 4, and contributed to the implementation of DRN (Section 4.7).

**Annette Möller** implemented the AR-EMOS method (Section 4.5).

**Cristina Primo** coordinated the verification work and the writing of Section 3.2.

**Gavin Evans and Ben Hooper** utilized the IMPROVER codebase to implement the Reliability Calibration method, along with bias correction approaches (Section 4.2).

**Markus Dabernig** participated in the collection of the data, and provided EMOS corrected forecasts (Section 4.4).

**Bert Van Schaeybroeck** contributing to the verification results (Section 3.2).

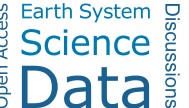

**David Jobst** implemented the DVQR method (Section 4.6).

**Aitor Atencia** contributed to the verification work.

**Jonas Bhend** participated in the collection of the data, contributed to the verification work by producing Figs. 3-5 and drafted the results Section 5.

**Zied Ben Bouallègue** implemented the ASRE postprocessing method (Section 4.1).

**Stéphane Vannitsem** leads the PP module of EUMETNET within which the Benchmark has been conceptualized.

**Peter Mlakar** developed and implemented the ANET method (Section 4.8).

**Janko Merše and Jana Faganeli Pucer** contributed to the development and implementation of the ANET method (Section 4.8).

**Olivier Mestre and Maxime Taillardat** participated in the collection of the data.

All the authors participated to the writing and the review of the manuscript.

*Competing interests.* The authors declare no competing interests.

*Acknowledgements.* Jonathan Demaeyer thanks Florian Pinault and Baudoin Raoult from ECMWF for their help on the setup of the climetlab plugin, and Francesco Ragone, Lesley De Cruz and David Docquier from RMIB for their support to gather the gridded the data. He also thanks Veerle De Bock and Joffrey Schmitz for their help with the RMIB data. The authors thank Tom Hamill for his guidance on the selection of variables during the dataset design phase. The EUMETNET PP module thanks the ECMWF for its support on the European Weather Cloud. Jieyu Chen, Nina Horat, and Sebastian Lerch gratefully acknowledge support by the Vector Stiftung through the Young Investigator Group "Artificial Intelligence for Probabilistic Weather Forecasting". Annette Möller acknowledges support by Deutsche Forschungsgemeinschaft (DFG) Grant Number MO 3394/1-1, by the Hungarian National Research, Development and Innovation Office under Grant Number NN125679, and by the Helmholtz Association's pilot project "Uncertainty Quantification".



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
