# Peer review of "The EUPPBench postprocessing benchmark dataset v1.0"

_Earth System Science Data, 2022_

## Author Response (AR1)

Response to Referee

Referee #1

**Referee's comment**

*This paper introduces a benchmark dataset that can be used to compare the performance of statistical postprocessing methods for several key weather variables. Such a dataset is very useful to the postprocessing community in which numerous methods have been (and continue to be) developed but are often difficult to rank because they have been tested in different setups. While particular setups often require tailored solutions, at least subgroups of postprocessing methods can be considered to have interchangeable areas of application, and a benchmark data set can thus be useful to identify methods that stand out as particularly powerful. Along with the introduction of the data set the authors use the example of 2 metre temperatures to demonstrate how such an intercomparison of methods could be performed. I welcome this initiative, and I only have one minor comment and a few suggestions regarding language.*

*Minor comment:*

*Section 4.8: Is there a reference for this ANET method? If a reference is provided, the description of the method is adequate. Otherwise, more explanation is required for details like the dynamic attention mechanism. Since methodology is not the focus of this paper, such information could be provided as supplemental material.*

*Typos and language:*

*60: complement -> complemented*

*113: as aforementioned -> as mentioned before*

*198: significant tests are run -> statistical tests are performed*

*216: validity -> valid*

*345: 'as' seems unnecessary here*

**Our response**

We thank the referee for his/her nice and encouraging review. We have address his/her minor comment as follow: A reference to a preprint (https://arxiv.org/abs/2303.17610) by Peter Mlakar and co-authors has been added, which provides many details on the ANET method. In addition, the typos and language issues have been addressed.

Referee #2

**Referee's comment**

*This paper introduces a publicly available dataset meant to help standardize and compare studies of various methods for postprocessing ensemble forecasts. The paper is welcome because of the substantial ongoing activity in developing such methods. Much of this recent activity has involved machine learning methods, and I personally have noticed that very few such studies adequately (including computation of possibly statistically significant differences) compare their results to the more conventional, established methods. The dataset described here should facilitate such comparisons, to the benefit of the discipline and quality improvements in weather forecasts. The paper should be accepted after consideration of the following comments:*

*The paper would benefit from copy-editing by a native English speaker, to correct the grammatical errors that occur in various parts of the paper.*

*The caption for Fig. 1 should include an explanation of the various rectangular regions.*

*Line 184. Disagree that bias characterizes skill of anything. "Skill" refers to relative accuracy, and bias is not a measure of accuracy. For example, unbiased forecasts can result from equal numbers of extreme (and therefore poorly skilled) overforecasting and underforecasting.*

*Line 186. Equality of average ensemble standard deviation and RMS error of the ensemble mean is indicative of calibration only if the forecasts are unbiased, because (squared) bias will inflate RMS error but will not affect ensemble standard deviation. This metric therefore cannot distinguish bias from ensemble underdispersion. Rank histograms or reliability diagrams are better indicators of calibration. To what extent do the nonzero biases in Fig 3b affect the spread-error-ratio results in Fig 3d? Similarly, in Figure 6 (line 335ff).*

*Equation 1. The overbar notation usually denotes sample average. Perhaps use a "prime" accent instead to denote the anomalies.*

*line 247. Should specify that the preserved correlations are simultaneously spatial, temporal, and intervariable.*

*Section 5. It would be worthwhile to give at least an indication of the sampling variability of the presented verification statistics, if not full hypothesis test results, in order to give the reader an idea of the magnitudes of differences that are meaningful. In particular, I am interested to see whether the substantially greater computational overhead of the machine learning methods results in significantly better performance. Line 322 notes that "most postprocessing methods perform similarly".*

**Our response**

We thank the referee for his/her nice and thorough review. We have addressed his/her comments as follow:

1. A native English speaker has reviewed the paper for grammar and spelling. Thank you for the suggestion.
2. **Caption for Fig. 1:** There is now a description of the rectangular area shown on Figure 1 in the caption:
   - "Spatial coverage of the dataset. Blue rectangle: spatial domain of the gridded dataset. Blue dots: position of the stations included in the dataset. Grey lines depict the latitude and longitude grid."
3. **Line 184:** This sentence has been replaced by the following one (now **Line 189-190**):

- "The Bias is defined as the average difference between the ensemble mean and observation, and points out if an ensemble has positive or negative systematic errors."

4. **Line 186:** We also computed the spread to standard deviation of errors ratio which is not affected by the biases and compared it to the spread-to-error ratio. The differences for all but ECMWF DMO are rather small (barely noticeable). While the issue of systematic biases affecting the spread-to-error (s2e) ratio is indeed present, the s2e is in our view more indicative of deficiencies in the postprocessing than the spread to standard deviation of errors. Hence our choice to display it. We are now commenting on this issue in the manuscript (**Line 358-362**):

   - "Please note that the strong under-dispersion of ECMWF DMO as indicated by the spread-error-ratio in Fig. 3 and Fig. 6 is slightly reduced when systematic biases are removed (not shown). For the postprocessed forecasts, the effect of remaining systematic biases on the spread-error-ratio is negligible. More detailed analyses of the results would of course be possible, but are beyond the scope of this publication on the benchmark dataset, and will be the subject of a dedicated work. "

5. **Equation 1:** We have changed the notation as suggested by the referee. Thank you for pointing that out.

6. **Line 247:** Indeed, thank you for pointing that out. As suggested, we now state (**Line 267-268**):

   - " One of the main advantages of MBM postprocessing is that - by design - it

     preserves simultaneously spatial, temporal, and inter-variable correlations in the forecasts."

7. **Section 5:** Thank you for this suggestion. We have added an appendix section A detailing the results of the comparative significance testing (on the CRPS statistics). Regarding its relationship with the computational cost, we consider that it is beyond the scope of the present study. Indeed, comparing the computational ressources needed by different postprocessing methods is a very difficult challenge here. Some of our co-authors have already performed such kind of comparison in past studies, and pointed out that due to the many different contributors and machines involved here, it is a too complicated task for the simple benchmark that we want to present in this article. Nonetheless, we have provided some logs of the resources and time used in the archive of the models output available on Zenodo. This should allow the reader to make up his mind about this issue. We also now mention the challenge of doing such comparison in the article (**Line 226-230**):

   - "Note that a direct comparison of computational costs is challenging because of the differences in terms of the utilized hardware infrastructure, software packages and parallelization capabilities, and might be considered in future work, ideally within a fully automated procedure (see Section 6).

     That said, the computational costs of all considered post-processing methods are by several orders of magnitude lower than those required for the generation of the raw ensemble forecasts."

**Referee #3**

**Referee's comment**

*The paper introduces the first version of the EUPPBench benchmark dataset for statistical post-processing, which contains both gridded and station data of several weather quantities and different constant fields as well. The idea behind the initiative is very clear as this analysis-ready dataset provides a standardized tool for testing new approaches to post-processing and comparing their forecast skill to the existing well established methods. Besides introducing the dataset itself, the authors provide a very useful example together with the corresponding R and Python codes. They calibrate 2m temperature forecasts with the help of the state-of-the-art post-processing methods ranging from simple accounting for systematic and representativeness errors and reliability calibration to highly advanced machine learning-based distributional regression networks and ANET. The paper is well written, but far from being self-contained, one has to follow the provided references to understand completely the considered methods. However, this is not a disadvantage, as the main goal is to demonstrate the utility of the benchmark dataset. I have just a few minor remarks to be addressed.*

1. *P1L13: "learn about the models weather forecast deficiencies": please reformulate.*
2. *P2L23: and P24L503: Instead of the arXiv preprint of Lakatos et al. 2022, please cite the published one (Lakatos et al. 2023)*

   *Lakatos, M., Lerch, S., Hemri, S. and Baran, S.: Comparison of multivariate post-processing methods using global ECMWF ensemble forecasts, Quarterly Journal of the Royal Meteorological Society, https://doi.org/10.1002/qj.4436, 2023.*
3. *P15L93: please provide some reference to ANET or a more detailed description.*
4. *Following the one shot installation process of the codes suggested in https://github.com/EUPP-benchmark/ESSD-benchmark, I was not able to download code for ANET. Please check.*

**Our response**

We thank the referee for his/her nice and constructive review of the manuscript. We have addressed his/her minor comments in the following way:

1. We have reformulated the sentence as (**Line 12-13**):

   - "The term "postprocessing techniques" here refers to methods which use past forecasts and observations to learn information about forecast deficiencies, that can then be used to correct future forecasts."
2. The reference has been updated. Thank you for pointing that out.
3. A reference to a preprint (https://arxiv.org/abs/2303.17610) by Peter Mlakar and co-authors has been added, which provides many details on the ANET method.
4. The problem of the installation process has been identified and fixed. Thank you for raising that issue.